# Fitness and Hard Seededness of F_2_ and F_3_ Descendants of Hybridization between Herbicide-Resistant *Glycine max* and *G. soja*

**DOI:** 10.3390/plants12213671

**Published:** 2023-10-25

**Authors:** Rong Liang, Jia-Li Liu, Xue-Qin Ji, Kenneth M. Olsen, Sheng Qiang, Xiao-Ling Song

**Affiliations:** 1Weed Research Laboratory, College of Life Sciences, Nanjing Agricultural University, Nanjing 210095, China; 2020216029@stu.njau.edu.cn (R.L.); 2021116017@stu.njau.edu.cn (J.-L.L.); 2020116016@stu.njau.edu.cn (X.-Q.J.); wrl@njau.edu.cn (S.Q.); 2Department of Biology, Washington University in St. Louis, St. Louis, MO 63130, USA; kolsen@wustl.edu

**Keywords:** wild soybean (*Glycine soja*), transgenic soybean, plant invasion, weed management, seed bank

## Abstract

The commercial cultivation of herbicide-resistant (HR) transgenic soybeans (*Glycine max* L. Merr.) raises great concern that transgenes may introgress into wild soybeans (*Glycine soja* Sieb. et Zucc.) via pollen-mediated gene flow, which could increase the ecological risks of transgenic weed populations and threaten the genetic diversity of wild soybean. To assess the fitness of hybrids derived from transgenic HR soybean and wild soybean, the F_2_ and F_3_ descendants of crosses of the HR soybean line T14R1251-70 and two wild soybeans (LNTL and JLBC, which were collected from LiaoNing TieLing and JiLin BaiCheng, respectively), were planted along with their parents in wasteland or farmland soil, with or without weed competition. The fitness of F_2_ and F_3_ was significantly increased compared to the wild soybeans under all test conditions, and they also showed a greater competitive ability against weeds. Seeds produced by F_2_ and F_3_ were superficially similar to wild soybeans in having a hard seed coat; however, closer morphological examination revealed that the hard-seededness was lower due to the seed coat structure, specifically the presence of thicker hourglass cells in seed coat layers and lower Ca content in palisade epidermis. Hybrid descendants containing the *cp4-epsps* HR allele were able to complete their life cycle and produce a large number of seeds in the test conditions, which suggests that they would be able to survive in the soil beyond a single growing season, germinate, and grow under suitable conditions. Our findings indicate that the hybrid descendants of HR soybean and wild soybean may pose potential ecological risks in regions of soybean cultivation where wild soybean occurs.

## 1. Introduction

Genetically modified (GM) soybean (*Glycine max* Linn. Merr.) is one of the world’s four most widely cultivated GM crops, with a planting area that increased from 500,000 hectares in 1996 to 91.9 million hectares in 2019, accounting for 48% of the global GM crop planting area [1,2]. Among GM soybean traits, herbicide resistance (HR) is the most important. In China, three of the four safety certificates for GM soybeans issued by the Ministry of Agriculture and Rural Affairs are for varieties with HR traits. Soybean’s domestication origin is in eastern Asia, and one of the possible ecological risks posed by the commercial cultivation of HR soybeans in China is the potential for pollen-mediated gene flow to wild soybeans (*Glycine soja* Sieb. et Zucc.), creating GM hybrids whose descendants could persist indefinitely in the wild.

Wild soybean is the direct ancestor of cultivated soybean, and both *Glycine* species have the same chromosome number (2n = 40). Wild soybean, which occurs in all of China and neighboring regions of eastern Asia, is of great value in studying the origin and evolution of soybean [3,4,5]. While both species are predominantly self-pollinating, occasional cross-pollination can lead to gene flow from cultivated soybean to wild soybean since there is no reproductive isolation between them [5,6,7,8]. Pollen flow and hybridization between HR soybean and wild soybean have been widely reported [4,9,10,11]. However, whether HR soybean genes can introgress into the wild population also depends on the fitness of hybrids and descendants. Fitness is considered to value the adaptation of individuals or populations with specific genotypes under different ecological conditions [12]. Wild soybean is characterized by high genetic diversity [3,13], which could make the hybrids difficult to control, and the hybrids could potentially contaminate wild germplasm resources. Therefore, before introducing widespread commercial planting of HR soybean in regions where wild soybean occurs, it is important to evaluate the multigenerational fitness of hybrid descendants resulting from gene flow from HR soybeans to wild soybeans.

Seeds of wild soybean are characterized by a hard, impermeable seed coat, a trait referred to as hardseededness [5,14,15]. Hard seededness is one of the dormancy traits of wild soybean that inhibit germination until favorable conditions appear [16]. In contrast, the seeds of cultivated soybean are protein rich and perishable, which prevents the domesticated species from overwintering and persisting outside of cultivation [17]. Previous studies have established that the hybrids of wild soybean (as the seed parent) and HR soybean (as the pollen donor) were more similar to wild soybeans in seed morphology due to the segregation distortion, and the hard seed coat of hybrids needed to be scarified to break dormancy [14,18]. Studies have also shown that hybrids can complete the entire life cycle and that their fitness in soybean fields is comparable to or higher than that of wild soybean [19,20,21]. However, it is unknown to what extent crop-wild hybrids and their descendants consistently show the hard seededness that would be required for survival and long-term persistence outside of cultivation.

In a previous work, we evaluated the sexual compatibility of 10 populations of wild soybean with HR soybeans [22]; for 9 of the F_1_ created, we determined that the fitness of hybrids was significantly lower than that of the corresponding wild soybean parent [14]. However, that study did not examine fitness past the F_1_. In order to further explore the continuous impact of HR soybean gene flow on wild populations and the environment, the fitness of F_2_ and F_3_ derived from crosses of HR soybean line T14R1251-70 and two wild soybeans, LNTL and JLBC, which were collected from LiaoNing TieLing and JiLin BaiCheng, respectively, was investigated under two soil conditions and with or without weed competition in the current study. In addition, the hard seededness of hybrid seeds was assessed by observing the seed coat structure and determining the emergence rate after burying in different soil depths for different lengths of time. Our results on seed hardness and fitness of the F_2_ and F_3_ suggest that cultivation of HR soybean may pose risks for transgene escape to wild soybean and persistence of crop-wild hybrid descendants in the wild.

## 2. Results

### 2.1. Emergence Rate

For JLBC F_2_, the mean emergence rate was 90.8%, which was significantly higher than the mean value of its wild parent grown in the same experiment (79.2%) (*p* < 0.05); no significant difference in mean emergence rates was observed for JLBC F_3_ compared to its wild parent (Figure 1). In contrast, the mean emergence rate value of LNTL F_2_ (77.5%) was significantly lower than that of its wild parent (88.1%) (*p* < 0.01); for LNTL F_3_ and its wild parent, no significant difference was observed. Thus, variation in emergence rate differed in opposite directions at the F_2_ between the two wild populations, and they were not consistent between the F_2_ and F_3_ generations for either population.

### 2.2. True Leaf and Cotyledon Size

For the JLBC F_2_ and F_3_, both generations had statistically greater mean values of true leaf length than their wild soybean parent (15.8% and 13.2% longer, respectively). For JLBC F_3_ only, true leaf width was statistically smaller than JLBC. Similarly, for the LNTL, F_2_ and F_3_ had significantly greater mean values of true leaf length compared to the LNTL (7.79% and 29.7% longer, respectively). However, the mean leaf width of the LNTL F_3_ was also significantly greater than that of the wild parent.

No clear pattern was apparent for cotyledon size data. JLBC F_3_ were significantly smaller than those of JLBC. Mean cotyledon width of LNTL F_2_ was significantly smaller than that of LNTL, whereas for LNTL F_3_, the mean values of both cotyledon length and width were significantly greater than those of their wild soybean parent (Figure 2).

### 2.3. Plant Height at Third Trifoliolate Leaf Stage

There were differences in the mean plant height of JLBC, JLBC F_2_, and F_3_ among the four planting conditions. The mean values of JLBC F_2_ and F_3_ were 3.9–11.7% higher than JLBC. There was no significant difference in plant height of LNTL or F_2_ among the four planting conditions, and the mean plant height of LNTL and F_3_ was significantly higher when pure planted in farmland soil than that in wasteland soil. Under the same planting conditions, the mean plant heights of F_2_ were 16.70–20.30% higher and F_3_ were 36.98–44.63% higher than those of LNTL, respectively (Figure 3).

### 2.4. Aboveground Dry Biomass

The mean aboveground dry biomass of JLBC F_2_ was higher than that of their wild parent under pure planting, and that of JLBC F_3_ was higher in farmland soil. JLBC F_2_ and F_3_ had 1.99–3.71 times greater mean aboveground dry biomass than JLBC under the same planting condition. The mean aboveground dry biomass of LNTL F_2_ under mixed planting in farmland soil was significantly higher than that under the other three conditions, and that of LNTL in the same year was not significantly different among planting conditions. The aboveground dry biomass of LNTL F_3_ and LNTL was significantly higher under pure planting in farmland soil and significantly lower under mixed planting in farmland soil than those under the other two conditions. Under the same planting conditions, the mean aboveground dry biomass of LNTL F_2_ and F_3_ was significantly higher than that of LNTL; F_2_ was 1.3–1.59 times higher, while F_3_ was 1.59–1.77 times higher than LNTL (Figure 4A,B).

Aboveground dry biomass of weeds in farmland soil was always higher than that in wasteland soil (Figure 4C,D). There were no significant differences between the weed biomass with JLBC F_2_, F_3_, and JLBC. There was no significant difference between the weed biomass with LNTL F_2_ and with LNTL, while that of LNTL F_3_ was significantly lower than LNTL in both farmland and wasteland soil.

### 2.5. Vitro Pollen Germination Rate

The pollen germination rates of JLBC F_2_ were higher when pure planted than when mixed planted, while those of JLBC F_3_ were higher in farmland soil than in wasteland soil. The mean pollen germination rates of JLBC F_2_ were higher than or comparable to JLBC, and those of JLBC F_3_ were significantly lower than JLBC. The pollen germination rates of LNTL F_2_ and its wild soybean had the same trend under four conditions, with the highest under mixed planting in farmland soil or comparable. That of LNTL F_3_ and its wild soybean also had the same trend under four conditions, with the highest under pure planting in farmland soil and the lowest under mixed planting in wasteland soil. Under the same planting conditions, the pollen germination rate of LNTL F_2_ and F_3_ was 7.49–15.08% lower than that of LNTL (Figure 5).

### 2.6. Pod and Filled Seed Number per Plant

Mean values for pod and filled seed number per plant of JLBC F_2_ and JLBC were higher under pure planting than under mixed planting conditions, while mean values for JLBC F_3_ and JLBC were higher in farmland soil than in wasteland soil. Pod and filled seed numbers per plant for JLBC F_2_ and F_3_ were 1.1–3.7 times higher than JLBC in all four conditions.

Mean values for pod and filled seed number per plant of LNTL F_2_ and its wild soybean were significantly higher under mixed planting in farmland soil than in the other three planting conditions. In contrast, mean values for pod and filled seed number per plant of LNTL F_3_ and LNTL were significantly higher under pure planting in farmland soil than in the other three conditions. Under the same planting conditions, the mean number of pods per plant of LNTL F_2_ and F_3_ was 8.46–24.28% higher than that of LNTL (Figure 6).

### 2.7. 100-Seed Weight

The mean values of 100-seed weight for self-pollinated seeds of JLBC F_2_ and F_3_ were significantly lower than JLBC. JLBC F_3_ under mixed planting in farmland soil had significantly lower mean values than under other planting conditions; the values for JLBC F_2_ and F_3_ under other planting conditions were similar. Under the same planting condition, the mean 100-seed weight values for self-pollinated seeds of LNTL F_2_ and F_3_ were significantly higher than those of LNTL, with mean values 1.56–1.92 times greater than those of the wild parent. The mean 100-seed weight of LNTL F_3_ under pure planting in farmland soil was significantly higher than that of the other three conditions, while others were similar (Figure 7).

### 2.8. Relative Composite Fitness

Taking wild soybean as the standard “1”, the values of correspondingly F_2_ and F_3_ were valued as the relative composite fitness. The relative composite fitness of JLBC F_2_ and F_3_ was higher than that of JLBC under all four planting conditions, but not statistically significant. The relative composite fitness of JLBC F_2_ among four planting conditions had no difference, while that of JLBC F_3_ was higher under pure planting conditions or in farmland soil. The relative composite fitness of LNTL F_2_ and F_3_ was higher than that of LNTL under all four planting conditions, but the difference was not significant for F_2_, while it was significant for F_3_. There was no significant difference between LNTL F_2_ and its wild parent among the four conditions, and both LNTL F_3_ and LNTL had significantly higher fitness when pure planted in farmland soil than under the other three conditions, while there was no significant difference among the three conditions (Figure 8).

### 2.9. Hard Seededness and Germination Rate

Self-pollinated seeds of LNTL F_2_ and F_3_ were used to conduct this experiment. The hard seededness rate of LNTL F_2_ seeds was 89.50%, which was extremely significantly lower than that of LNTL (98.50%), and there was no significant difference between LNTL F_3_ seeds and LNTL seeds. After scarification, there was no significant difference in germination rate between F_2_/F_3_ seeds and wild soybean seeds (Table 1).

### 2.10. Seed Coat Structure

Self-pollinated seeds of LNTL F_2_ were used to conduct this experiment. There are obvious pits on the surface of the HR soybean seed coat, and the shape of the pits is irregular (Figure 9A,B). There are three main types of pits: Shallow long pits, deep round pits, shallow round pits, and some pits with a crack width of 0.1–0.3 μm. The surface of the HR soybean has almost no attachment, and the stratum corneum is directly exposed to the outside. Both LNTL and its LNTL F_2_ seed coat surface are covered by a thick layer of sediment, similar to a bulge at the basin margin, and the entire seed coat surface is honeycomb-shaped (Figure 9E,I); there are no cracks on the surface of wild soybean or F_2_ seed coat. At the hila of wild soybean and F_2_, there is a middle dent and several multiple irregular cracks on both sides of the dent, with a width of 3–20 μm (Figure 9F,J). At the same time, no honeycomb-like sediment attachment was observed around the hila of LNTL and F_2_ seeds.

The seed coat structure of HR soybean, LNTL, and LNTL F_2_ all contains four cell layers, followed by the palisade epidermis, hourglass cells, parenchyma, and aleurone layer; LNTL and F_2_ seeds also have a stratum corneum over the seed coat. Among them, the aleurone layer has monolayer cells, which are not easy to recognize with SEM (Figure 9C,D,G,H,K,L).

The palisade epidermis of LNTL wild soybean was comparable to that of the F_2_ seeds, and both were higher than that of HR soybean. The hard seededness rates of LNTL, F_2_, and HR soybean seeds decreased from 98.50%, 89.50%, and 1.00%, respectively. However, the proportion of palisade epidermis thickness in the seed coat decreases with the decrease in hard seededness rate. The thickness of hourglass cells and their proportion increased with the decrease in hardness rate. The parenchyma layers of LNTL and F_2_ seeds were significantly thinner than those of HR soybean (Figure 10).

### 2.11. Mineral Element in Seedcoat

Self-pollinated seeds of LNTL F_2_ were used to conduct this experiment. The content of Ca in the seed coat palisade epidermis of LNTL F_2_ seeds was significantly lower than that of LNTL; however, there was no significant difference for other mineral elements that were measured (Figure 11).

### 2.12. Seed Vitality in Soil

For self-pollinated seeds of both JLBC F_3_ and JLBC, under both 3 cm and 10 cm of soil, the trend of natural emergence rate increased with time, and the emergence rate with seed scarification hardly changed over time. The emergence rate of JLBC F_3_ seeds was higher with seed scarification and lower without seed scarification than JLBC, respectively (Figure 12A).

For self-pollinated seeds of both LNTL F_2_ and LNTL, under both 3 cm and 10 cm soil, the trend of natural emergence rate increased with time, and the natural emergence rate of LNTL F_2_ seeds after burying for 6 months was significantly higher than that of LNTL. After seed scarification, the trend of the natural emergence rate of both soybeans decreased with time, and the natural emergence rate of F_2_ seeds after burying for 3 months was significantly higher than that of LNTL (Figure 12B).

For self-pollinated seeds of LNTL F_3_ and LNTL, both under 3 cm and 10 cm of soil, the trend of natural emergence rate of both seeds increased with time and decreased after burying for 15 months. The natural emergence rate of LNTL F_3_ seeds after burying was higher than that of LNTL but not significantly. The emergence rate with scarification of LNTL F_3_ seeds was higher than that of LNTL but only significant at one time point (Figure 12C).

## 3. Discussion

### 3.1. Fitness of F_2_, F_3_ Compared with Parents

The F_2_ and F_3_ of this experiment were obtained by hybridizing HR soybeans (as the pollen donor) with wild soybeans (as the seed parent). The genetic difference between cultivated soybeans and wild soybeans derives from the domestication by humans of the wild species into the cultivated crop species [23,24,25,26]. In the domestication of legumes, selection favors enhanced aboveground traits, including greater seed size and palatability, reduced seed dormancy, and other desirable agronomic traits. The hybrids of cultivated soybean and wild soybean usually have a growth advantage over wild soybean [6,21]. In the context of crop improvement, hybridization of domestic soybeans (as the seed parent) and wild soybeans (as the pollen donor) can improve the resistance of hybrids and even promote the diversity of varieties [27,28]. However, if it is allowed to grow outside of cultivation, these same fitness advantages create potential ecological risks, particularly in regions of transgenic HR soybean cultivation; in this context, the advantages of hybrids do not bode well.

In our previous study, it was found that F_1_ hybrids of HR soybean and wild soybeans, including LNTL and JLBC, had lower fitness than the wild soybean parents [14]. In this study, F_2_ and F_3_ of both LNTL and JLBC, regardless of soil conditions and whether there was weed competition, showed significantly elevated mean values relative to their wild parents for multiple fitness-related traits, including plant height, number of pods per plant, number of filled seeds per plant, filled seed weight per plant, aboveground dry biomass, and 100-seed weight. The mean composite fitness of LNTL F_3_ under all four planting conditions was significantly higher than that of LNTL. As the generations increased, the adverse effects of hybridization are gradually eliminated through gene segregation and recombination [29,30]. The wild soybean LNTL and JLBC were collected at high latitudes, and in-field experiments were at lower latitudes. The fitness of wild soybeans was decreased due to the shorter photoperiod and other unsuitable environmental factors [31,32]. After receiving pollen of HR soybean adapted to the climate of the experimental location, F_2_ and F_3_ inherited adaptability to the local climate and environment, which ultimately led to the improvement of the survival competitiveness of the hybrid descendants.

It is worth noting that regardless of the planting conditions, LNTL F_2_ and F_3_ pollen viability was significantly higher than that of wild soybeans, while JLBC F_3_ had lower pollen viability. Pollen activity reflects the quality of pollen, affects seed formation, and is an important indicator for valuing reproductive ability [33]. The probability of interspecific hybridization and the fertility of hybrid descendants depend largely on the homology of the genomes and the degree of homology, which determines the possibility of pairing and recombination between the chromosomes of the parents [34]. HR soybean and wild soybean both belong to *Glycine*, and they have the same chromosome number (2n = 40), but there are differences in chromosome behavior and the division cycle of meiosis [5,35]. Therefore, in meiosis, hybrid descendants would have abnormal chromosome behavior. And different populations of wild soybean have varying degrees of chromosomal abnormalities, which also reflect the diversity of the germplasm resources of wild soybeans.

### 3.2. Effects of Soil Nutrition and Competition on Fitness of F_2_, F_3_

As the substrate for crops, there are various elements and substances that affect the development and reproduction of plants in the soil [36,37]. Unlike other crops, legumes have the ability to symbiotically fix nitrogen with nitrogen-fixing bacteria [38,39]. Therefore, the growth of soybean is not only affected by soil nutrition, especially nitrogen in the soil [40]. It was proven that the nitrogen-fixing capacity of cultivated soybean and wild soybean and the interaction mode with rhizosphere microorganisms are different [17,41,42].

In this experiment, soils from farmland and wasteland were used to plant the hybrids. The results showed that when there was no weed competition, except for LNTL F_2_, the fitness of JLBC F_2_, JLBC F_3_, LNTL F_3_, and their wild soybean under the farmland soil was significantly higher than that of the wasteland soil. The nitrogen form in soil may partly explain this anomalous difference in LNTL F_2_. Nitrate nitrogen and ammonium nitrogen, which are called available nitrogen, are effective forms of nitrogen nutrients in soil and can be directly absorbed and utilized by roots [43,44]. For JLBC F_3_ and LNTL F_2_, the content of available nitrogen was 10.71 mg/kg in wasteland soil and 23.59 mg/kg in farmland soil, both were not high enough for growth. At the same time, there was no significant difference in total nitrogen content between farmland soil and wasteland soil this year. This may explain the similar fitness of JLBC F_3_ and LNTL F_2_ and their wild soybeans in both wasteland and farmland soils. The restriction on the growth of JLBC F_3_ and LNTL F_2_ in wild soybean may be due to a lack of available phosphorus. Under limiting phosphorus, the uptake and utilization of nitrogen and other metabolic pathways will also be affected [45,46,47,48]. Therefore, when the nutrients were relatively abundant, the available nitrogen and available phosphorus, which were significantly different between farmland and wasteland soil, also had a significant impact on the growth of JLBC F_2_, LNTL F_3_, and wild soybean.

All hybrids have similar patterns in different soils to wild soybeans, suggesting that the utilization pattern of soil nutrients of hybrids and symbiotic nitrogen fixation are inherited from the seed producer, the wild soybean.

Weeds not only compete with crops for light [49,50] and nutrition in soil [51], but also change the environment and microorganisms of the rhizosphere through root exudates, which affects the growth of soybeans [52,53]. When there was weed competition, the number of pods and filled seeds per plant of JLBC F_3_, LNTL F_2_, and their wild soybean in farmland soil were significantly higher than those in wasteland soil, but the fitness of JLBC F_2_, LNTL F_3_ and their wild soybean was exactly opposite. This difference came from differences in the nutrient content of the soil used in the three-year trial. Weed dry biomass can reflect the nutrient level of the soil. It can also be seen that LNTL wild soybeans are less competitive with weeds than LNTL F_3_. This increased competitiveness may come from the genes of the paternal HR soybeans [54]. Although the available nitrogen level in wasteland soil was relatively low, the nitrogen fixation ability of hybrids and wild soybeans could still maintain the nitrogen balance in the soil and the normal growth of plants. The number of pods and filled seeds per plant under LNTL F_3_ mixed planting in wasteland soil was significantly higher than that of mixed planting in farmland soil. This phenomenon may also come from biological nitrification inhibition [55]. When there was weed competition, weeds, soybeans, soil nutrients, and the rhizosphere formed a complex interacting system [56,57,58,59]. The environment was changed to benefit the strong side, such as wild soybeans.

### 3.3. Seed Coat Structure and Seed Dormancy

Honeycomb epidermal attachments may be the first barrier to prevent the seed from absorbing water and expanding, and they are an important way for the seed to remain dormant. There is no obvious attachment on the surface of the seed coat of HR soybean, and the dormancy ability of crop seeds is almost completely lost. This attachment comes from the endocarp, known as bloom, and directly acts to change the gloss of the surface of the seed, reducing the chance that the seed will be found and eaten by animals [60,61]. Meanwhile, bloom has been proven to be related to seed oil content [62], and the change in soybean permeability in domestication was caused by human selection. The difference in bloom explained the difference in natural emergence rates between HR soybean and wild soybeans, but it still does not explain the difference between hybrid descendants and wild soybeans.

The emergence rate of seeds with scarification showed that there was no significant difference in seed viability between seeds of LNTL F_2_, F_3_, and wild soybean, which showed similar embryonic activity. The hard seededness of wild soybean ensures long-term seed dormancy. With time, buried seeds of all hybrids and wild soybeans were more likely to break dormancy, and embryonic activity decreased. Point mutations in *Gm*Hs1-1 cause the loss of hard seededness and this gene correlates with the content of calcium in the seed coat [63]. In the experiment, the calcium content in the seed coat of LNTL F_2_ seeds was significantly lower than that of wild soybean, indicating that LNTL F_2_ seeds partially inherited the soft seed coat of HR soybean, resulting in its hard seededness being weaker than wild soybean. However, some soybeans promote water absorption and break dormancy while maintaining the calcium content of the seed coat by cracking through the seed coat. This is the case with irregular cracks on the surface of HR soybeans seeds observed by SEM, but LNTL F_2_ seeds did not have this character. The formation of such cracks may come from changes in the seed coat layers. The shape and number of hourglass cells are often thought to be strongly related to seed dormancy and viability [64,65]. Palisade epidermis and parenchyma of LNTL F_2_ seeds were both similar to those of wild soybean, but the hourglass cells were significantly higher than those of wild soybean and lower than those of HR soybean. Hourglass cells are associated with the accumulation of various enzymes associated with water absorption and germination, such as catalase [66,67]. The difference in the hourglass cell layer could exactly explain the decline in hard seededness rate of LNTL F_2_ seeds compared to wild soybean.

Therefore, the hybrid seeds of wild soybean and HR soybean reduced the hard seededness compared to wild soybeans through the thickening of hourglass cells and the reduction of calcium content in the palisade epidermis.

## 4. Materials and Methods

Herbicide-resistant transgenic soybeans T14R1251-70 were provided by the National Soybean Improvement Center of Nanjing Agricultural University. The HR soybean, containing one single-copy *cp4-epsps*, was obtained by Agrobacterium-mediated co-transformation of the receptor soybean NJR44-1, which is an elite line bred by the National Soybean Improvement Center of Nanjing Agricultural University. The HR soybean withstands 3600 g a.i. ha-1 41% glyphosate isopropylammonium AS (Roundup Ultra; Monsanto, St. Louis, MO, USA). Wild soybean populations were collected from Tieling, Liaoning Province, and Baicheng, Jilin Province. Crossed seeds were obtained by artificial hybridization of wild soybeans as the seed producer and HR soybeans as the pollen donor from 2016 to 2017 [22]. The crossed seeds were harvested from different seed producers individually and then stored at 4 °C until further use. Experiments were conducted in a greenhouse and net house at the Pailou Experimental Farm (32°20′ N, 118°37′ E), Nanjing Agricultural University, China, from 2018 to 2020.

### 4.1. Seed Treatment and Seeding

Scarify the seed coat of wild soybeans and hybrid descendants. Seeds were sown in a plastic cup with a hole at the bottom (a diameter of 7 cm and a height of 7.5 cm). The substrate for seeding was farmland soil and wasteland soil, as described in Table 2. Seedings were placed in a net chamber for normal water management, and all test materials were randomly placed in the net chamber and cultured under natural light and photoperiod, during which the temperature fluctuated between 20 and 38 °C.

### 4.2. Emergence Rate and Cotyledon, True Leaf Size

When the cotyledons of the plants are unearthed and completely green (about 2 weeks after sowing), the number of seedlings of soybean plants is counted. When the first compound leaf of the plant has formed and the leaves are wrinkled but not fully expanded, the longest and widest cotyledons and true leaves are determined using Vernier calipers; each single plant is a replicate, and 20 plants per material are randomly selected for measurement.

### 4.3. cp4-Epsps in Hybrids

After the first ternately compound leaf of the plant was unfolded, the *cp4-epsps* was detected by PCR with a specific primer (5′-GGCACAAGGGATACAAACC-3′; 5′-ACCGCCGAACATGAAGGAC-3′). Count the number of plants carrying resistance genes and plants without resistance genes, and use the chi-square test to verify whether the resistance of hybrid separation ratio results conform to Mendel’s law of 3:1. The specific formula is as follows:
(1)$$\chi^{2}=\frac{{\left[|b\times A_{1}-a\times A_{2}|-(a+b)/2\right]}^{2}}{a\times (A_{1}+A_{2})}$$


*χ*^2^ represents the chi-square value, such as *χ*^2^ < 3.84, that is, *p* > 0.05, indicating that the inheritance law of resistance genes in hybrids conforms to Mendel’s law of inheritance; *A*_1_ indicates the number of plants carrying resistance genes; *A*_2_ indicates the number of plants that do not carry resistance genes; F_2_: *a* = 3, *b* = 1; F_3_: *a* = 5, *b* = 1.

### 4.4. Planting Conditions

Wasteland soil and farmland soil were collected at the Pailou base. Take three copies of the soil and entrust Nanjing Zhongding Biological Company to test the physical and chemical properties of the soil (Table 2).

Four planting conditions were set: Pure planting in wasteland soil (PW), pure planting in farmland soil (PF), mixed planting with weeds in wasteland soil (MW), and mixed planting with weeds in farmland (MF). For the emergence rate test, 60 plants with consistent growth of HR soybean, LNTL, JLBC, and hybrid descendants were selected, and 15 plants were transplanted under four planting conditions. Under single planting conditions, a pod (23 cm in diameter and 25 cm in height) with bamboo was set for LNTL, JLBC, and hybrid descendants growth. When mixed planting with weeds, *Setariaviridis* (L.) *Beauv*. 0.5 g, *Digitariasanguinalis* (L.) Scop. 0.5 g, *Echinochloacolona* (L.) Link. 0.5 g, and *Eleusine indica* (L.) Gaertn. 0.25 g were sown evenly in pots (52 cm diameter and 35 cm height).

### 4.5. Fitness Determination

Investigate fitness indicators during plant vegetative and reproductive periods. Emergence rate: Two weeks after sowing, count the number of all seedlings unearthed with green cotyledons; true leaf size: When the first compound leaf has been formed and the leaf is wrinkled but not fully expanded, the cotyledon length width and true leaf length width are measured by vernier calipers; plant height: At the third-ternately-compound stage, the length from the tip of the main stem to the cotyledon ring was measured; pollen vitality: Randomly collect flower buds on plants at full bloom period (flag petals are 1–2 mm higher than sepals), culture in vitro for 60 min, and count the number of germinated pollen under a microscope; aboveground dry biomass: After harvesting, the aboveground part of the plant is dried to a constant weight and weighed; number of pods per plant: After harvesting, the total number of pods per plant is counted and artificially threshed; number of filled seeds per plant: After harvesting, select the filled seeds from all single seeds (with regular shape, no depression, and no shrink), count the number, and weigh them; composite fitness: Taking wild soybean as the standard “1”, the seedling emergence rate, cotyledon length × cotyledon width + true leaf length × true leaf width, plant height, aboveground dry biomass, pollen germination rate of 60 min, number of pods and filled seeds per plant, 100-seed weight to wild soybean were valued, and the composite relative fitness is the average of the values.

### 4.6. Seed Hard Seededness Rate and Scarified Emergence Rate

Fifty seeds were randomly selected from all the plants under the pure planting in soil with 4 repeats. The number of seeds that did not swell (seed size did not change) after 7 days of soaking in distilled water was counted. Hard seededness rate = number of unswollen seeds/total number of seeds×100%. After the hard seededness rate is determined, scarify the seed coat of the remaining hard wild soybean and hybrid seed without damaging the embryo. Incubate the scarified seeds at a constant temperature of 25 °C for 7 days; count the germinating seeds with a radicle length twice that of the seed length. Emergence rate (%) = total number of germinated seeds / total number of seeds × 100%.

### 4.7. Seed Coat Structure and Elemental Content

Select 3 filled seeds with a complete seed coat from plants purely planted in farmland soil. Cut the seeds along the seed ridge corresponding to the center point of the seed hilum to avoid damage to the embryo. Stick the cut seeds on the sample stage; use a Hitachi-1010 ion sputterer to spray gold on the surface; use a Hitachi-SU8010 scanning electron microscope for observation and photography; and use an SEM accelerating voltage of 20 kV. Photoshop (version 21.1.2; Adobe Systems Incorporated, San Jose, CA, USA) was used to measure the thickness of each structure. The elemental content of the palisade layer of the seed coat was determined with an X-ray spectrometer (HORIBA).

### 4.8. The Seed Vitality under Soil

Eighty seeds were randomly selected from each of the 15 plants purely planted in farmland soil, and they were packed into nylon mesh bags with a 0.2 mm pore size and buried in the research base of Nanjing Agricultural University in December of that year, 3 cm and 10 cm deep from the soil surface. The number of seeds that had emergence, the emergence rate after scarifying the seed coat, and the number of ungerminated seeds checked for rot and mildew were recorded.

### 4.9. Data Analysis

All data are statistically analyzed using SPSS (SPSS 22.0). Duncan’s multiple range test in the univariate ANOVA test was used to analyze the differences in fitness indexes of the same material under four planting conditions, the thickness of different cell layers of transgenic soybean, wild soybean, and hybrid descendant seed coat, and the proportion of total thickness. The independent sample T test was used to analyze the differences in fitness indexes and composite fitness, in hard seededness rate and emergence rate after nicking hard seed coat, and in mineral element content between wild soybean and hybrid, and the data were plotted with Prism GraphPad.

## 5. Conclusions

The fitness of the F_2_ and F_3_ of herbicide-resistant transgenic soybean line T14R1251-70 and wild soybean LNTL and JLBC was significantly increased under farmland and wasteland soil conditions, as well as with or without weed competition, and the competitiveness was significantly enhanced. Self-pollinated seeds produced by hybrid descendants were similar to wild soybeans with a hard seed coat but had a lower hard seededness rate due to the seed coat structure. The decrease in hard seededness was due to the thicker hourglass cells and the lower Ca content in the seed coat. Hybrid descendants containing modified gene *cp4-epsps* can complete life histories and produce a large number of seeds, which can persist in the soil for a long time, germinate, and grow under suitable conditions. So, the hybrid descendants of herbicide-resistant transgenic soybean and wild soybean have potential ecological risks.

## Figures and Tables

**Figure 1 plants-12-03671-f001:**
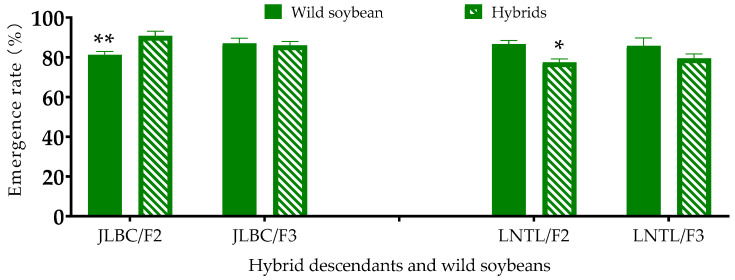
Emergence rate of F_2_, F_3_ and its wild soybean JLBC and LNTL. Note: * and ** indicates significant difference (*p* < 0.05) and extremely significant difference (*p* < 0.01) between hybrid descendants and its wild soybean.

**Figure 2 plants-12-03671-f002:**
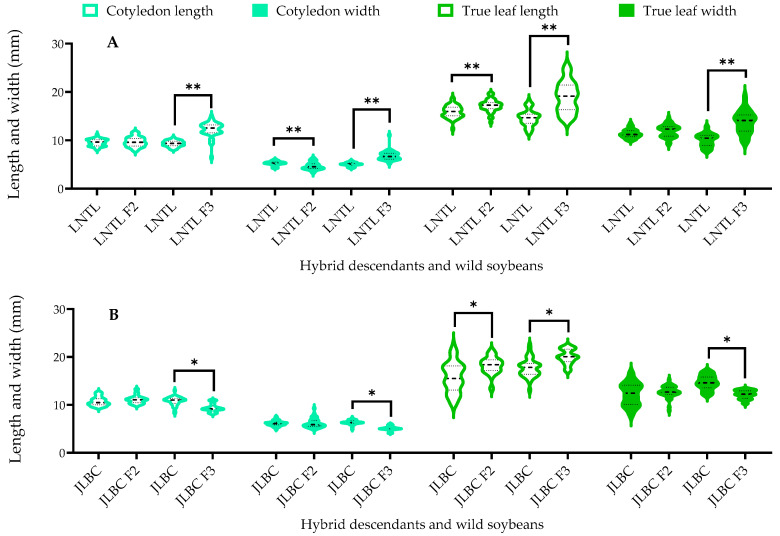
Size of cotyledon and true leaf of F_2_, F_3_ and wild soybeans ((**A**): LNTL; (**B**): JLBC). Note: * and ** indicates significant difference (*p* < 0.05) and extremely significant difference (*p* < 0.01) of the same trait between hybrid descendants and its wild soybean.

**Figure 3 plants-12-03671-f003:**
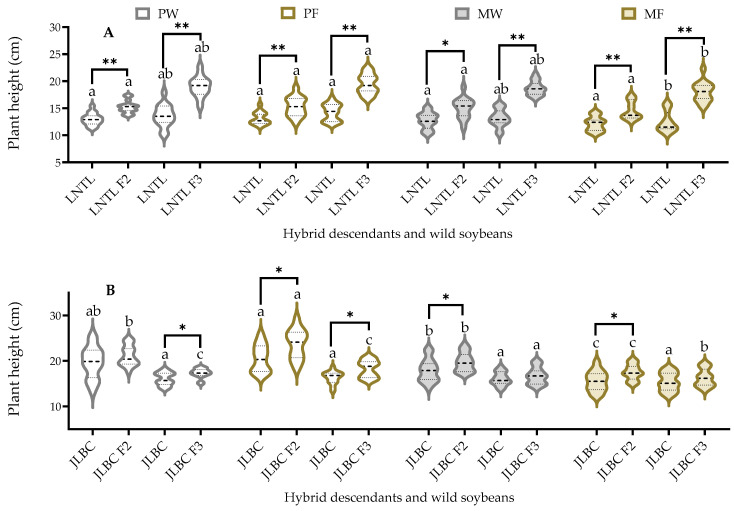
Plant height of F_2_, F_3_, and wild soybean ((**A**): LNTL; (**B**): JLBC) under four planting conditions (the third trifoliolate leaf stage). Note: PW: pure planting in wasteland soil; PF: pure planting in farmland soil; MW: mixed planting with weeds in wasteland soil; MF: mixed planting in farmland; * and ** indicates significant difference (*p* < 0.05) and extremely significant difference (*p* < 0.01) between hybrid descendants and its wild soybean. Different lowercase letters indicate significant difference (*p* < 0.05) of hybrid descendants or wild soybeans among four planting conditions.

**Figure 4 plants-12-03671-f004:**
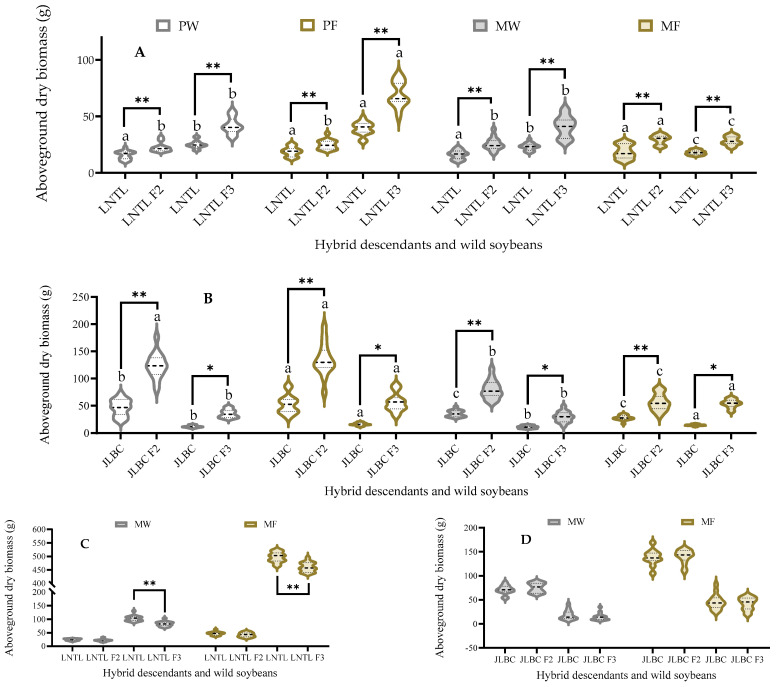
Aboveground dry biomass of F_2_, F_3_, wild soybeans (**A**,**B**) and weeds (**C**,**D**) under four planting condition. Note: PW: pure planting in wasteland soil; PF: pure planting in farmland soil; MW: mixed planting with weeds in wasteland soil; MF: mixed planting in farmland; * and ** indicates significant difference (*p* < 0.05) and extremely significant difference (*p* < 0.01) of the same trait between hybrid descendants and its wild soybean. Different lowercase letters indicate significant difference (*p* < 0.05) of hybrid descendants or wild soybeans among four planting conditions.

**Figure 5 plants-12-03671-f005:**
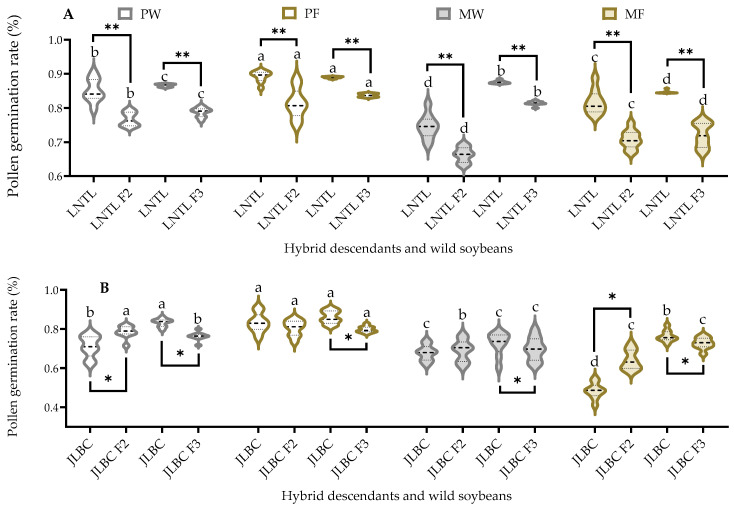
Vitro pollen germination rate of F_2_, F_3_, and wild soybeans ((**A**): LNTL; (**B**): JLBC) at 60 min under four planting condition. Note: PW: pure planting in wasteland soil; PF: pure planting in farmland soil; MW: mixed planting with weeds in wasteland soil; MF: mixed planting in farmland; * and ** indicates significant difference (*p* < 0.05) and extremely significant difference (*p* < 0.01) of the same trait between hybrid descendants and its wild soybean. Different lowercase letters indicate significant difference (*p* < 0.05) of hybrid descendants or wild soybeans among four planting conditions.

**Figure 6 plants-12-03671-f006:**
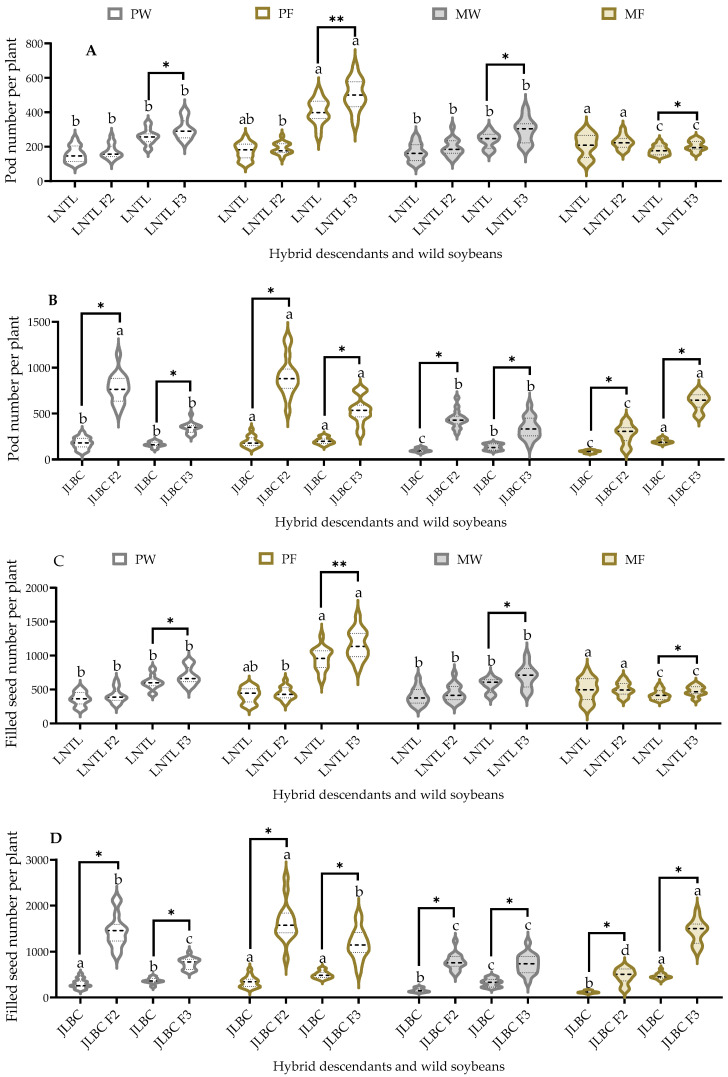
Pod number ((**A**): LNTL; (**B**): JLBC) and filled seed number ((**C**): LNTL; (**D**): JLBC) per plant of F_2_, F_3_ and wild soybeans under four planting condition. Note: PW: pure planting in wasteland soil; PF: pure planting in farmland soil; MW: mixed planting with weeds in wasteland soil; MF: mixed planting in farmland; * and ** indicates significant difference (*p* < 0.05) and extremely significant difference (*p* < 0.01) of the same trait between hybrid descendants and its wild soybean. Different lowercase letters indicate significant difference (*p* < 0.05) of hybrid descendants or wild soybeans among four planting conditions.

**Figure 7 plants-12-03671-f007:**
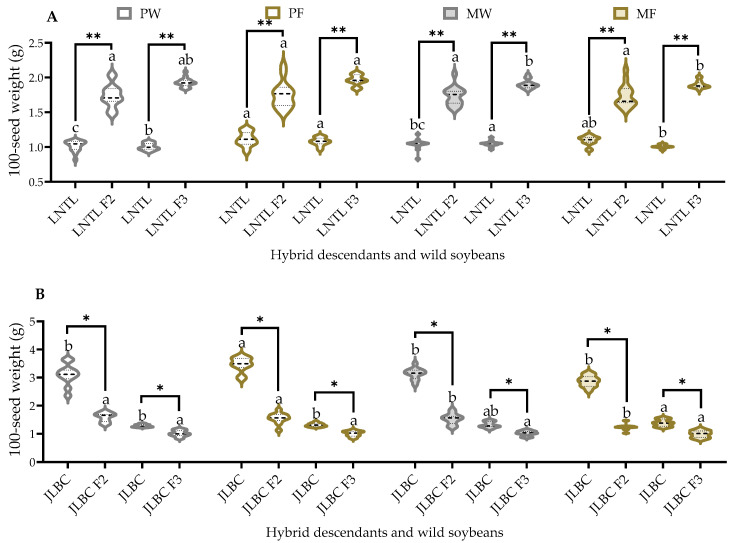
100-seed weight of seeds of F_2_, F_3_, and wild soybean ((**A**): LNTL; (**B**): JLBC) under four planting conditions. Note: PW: pure planting in wasteland soil; PF: pure planting in farmland soil; MW: mixed planting with weeds in wasteland soil; MF: mixed planting in farmland; * and ** indicates significant difference (*p* < 0.05) and extremely significant difference (*p* < 0.01) of the same trait between hybrid descendants and its wild soybean. Different lowercase letters indicate significant difference (*p* < 0.05) of hybrid descendants or wild soybeans among four planting conditions.

**Figure 8 plants-12-03671-f008:**
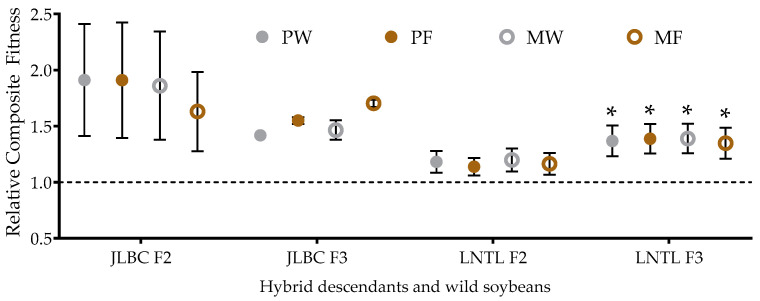
Comparison of composite fitness between wild soybeans and F_2_, F_3_. Note: PW: pure planting in wasteland soil; PF: pure planting in farmland soil; MW: mixed planting with weeds in wasteland soil; MF: mixed planting in farmland; The dashed line represents the composite fitness of wild soybean as 1, * indicates significant difference (*p* < 0.05) of the same trait between hybrid descendants and its wild soybean.

**Figure 9 plants-12-03671-f009:**
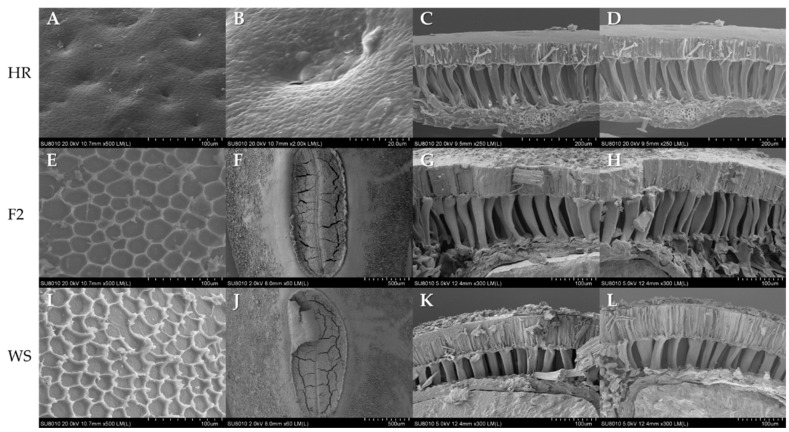
SEM of seed coat structure of transgenic soybean, LNTL wild soybean and F_2_ seeds. Note: (**A**) Seed coat surface of TS (×500); (**B**) depress and crack on seed coat surface of TS (×2000); (**C**,**D**) seed coat layers of TS (×250); (**E**) seed coat surface of LNTL F_2_ (×500); (**F**) hilum surface of LNTL F_2_ (×60); (**G**,**H**) seed coat layers of LNTL F_2_ (×300); (**I**) seed coat surface of LNTL F_2_ (×500); (**J**) hilum surface of LNTL (×60); (**K**,**L**) seed coat surface of LNTL (×300).

**Figure 10 plants-12-03671-f010:**
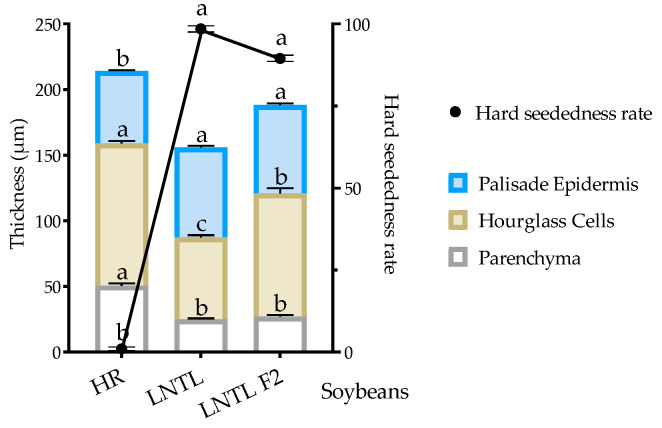
Relationship between hard seededness rate and thickness of seed coat layers of transgenic soybean, LNTL wild soybean and F_2_ seeds. Note: Different lowercase letters indicate significant difference (*p* < 0.05) among hybrid descendants, wild soybean, and transgenic soybean.

**Figure 11 plants-12-03671-f011:**
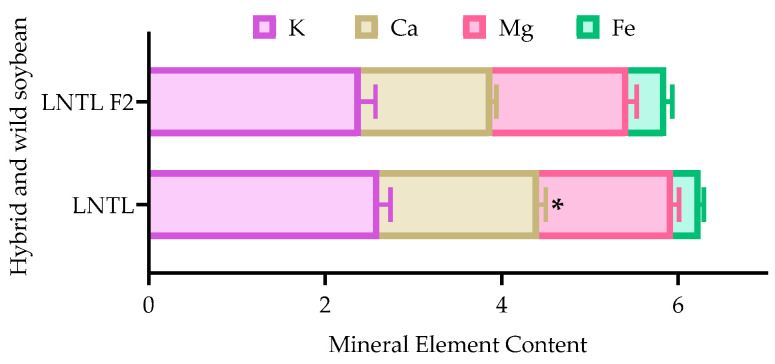
Main mineral element content of seed coat palisade epidermis of LNTL wild soybean and F_2_ seeds. Note: * indicates significant difference (*p* < 0.05) between hybrid descendants and wild soybean.

**Figure 12 plants-12-03671-f012:**
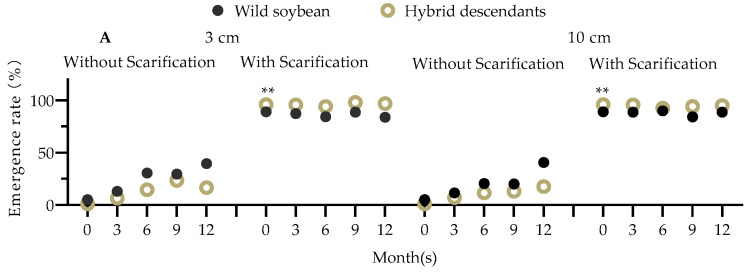
The emergence rate with or without scarification of seeds of JLBC F_2_ (**A**), LNTL F_2_ (**B**) and LNTL F_3_ (**C**) after burying in 3 cm or 10 cm soil. Note: * and ** indicates significant difference (*p* < 0.05) and extremely significant difference (*p* < 0.01) between hybrid descendants and its wild soybean.

**Table 1 plants-12-03671-t001:** Hard seededness rate and germination rate with scarification of LNTL F_2_, F_3_ seeds.

Material	Hard Seededness Rate (%)	Germination Rate with Seed Scarification (%)
LNTL	98.50 ± 0.96 **	94.44 ± 0.93
LNTL F_2_	89.50 ± 0.96	93.86 ± 0.53
LNTL	100	98.00 ± 0.00
LNTL F_3_	96.50 ± 0.02	94.50 ± 0.02

Note: ** indicates extremely significant difference (*p* < 0.01) between hybrids and its wild soybean.

**Table 2 plants-12-03671-t002:** Soil physicochemical properties per year.

	Soils	Organic Matter g/kg	Total Nitrogen g/kg	Total Phosphorus g/kg	Total Potassium g/kg	Available Phosphorus mg/kg	Available Nitrogen mg/kg
JLBC F_2_	Wasteland soil	2.79 ± 0.10	0.37 ± 0.01	0.56 ± 0.01	22.04 ± 0.46	22.39 ± 0.52	44.15 ± 0.2
	Farmland soil	38.51 ± 0.35 *	2.20 ± 0.03 *	1.76 ± 0.01 *	18.94 ± 0.19	47.81 ± 0.33 *	163.74 ± 0.54 *
JLBC F_3_ and LNTL F_2_	Wasteland soil	4.82 ± 0.22	0.27 ± 0.37	0.17 ± 0.11	9.79 ± 0.09	0.1 ± 0.03	10.71 ± 1.25
	Farmland soil	9.74 ± 0.81 *	0.37 ± 0.04	0.26 ± 0.12 *	10.07 ± 0.10	1.68 ± 0.31 *	23.59 ± 2.61 *
LNTL F_3_	Wasteland soil	7.78 ± 0.40	0.72 ± 0.02	0.25 ± 0.01	20.94 ± 0.42	9.99 ± 0.86	51.91 ± 1.38
	Farmland soil	11.19 ± 1.50	1.06 ± 0.11	0.36 ± 0.07	21.10 ± 0.48	28.21 ± 1.32 *	145.41 ± 21.08 *

Note: * indicates significant difference between wasteland soil and farmland soil (*p* < 0.05).

## Data Availability

Data is contained within the article. The data presented in this study are available in figures and tables.

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
