# Peer review of "Fitness and Hard Seededness of F2 and F3 Descendants of Hybridization between Herbicide-Resistant Glycine max and G. soja"

_plants, 2023, doi:10.3390/plants12213671_

Round 1

Reviewer 1 Report

This is an interesting manuscript, but you may improve this article in order to publish in this journal. Otherwise, I have a lot of recommendations to increase the quality of your paper. Be careful with the writing and mistakes.

I recommend to write the scientific names of the species in the article title because this is a botanical journal.

Line 10. When you write an acronym you must write in capitals the letters that you use to build it, for example, in the abstract you must write as follows: “Herbicide-Resitant (HR)”. This simple thing will make the reading easier for potential and future readers.

Lines 10-11. You must avoid the use of square brackets.

Line 28. There are several keywords repeated in the article title. The keywords are “Wild soybean”, “transgenic soybean” and “fitness”. In order to increase the visibility of your paper I recommend changing these keywords. If you change them by other keywords, you will increase the probability that your paper could be found by future readers when they look for your paper in some databases like Scopus for example. If you repeat the same words in the article title and in keywords, less people could find your work. So, you must think about the visibility of your research.

Line 15. You must write in brackets the meaning of each acronym in the text.

Lines 31. You must avoid the use of square brackets.

Line 31. When you write an acronym you must write in capitals the letters that you use to build it, for example, in the abstract you must write as follows: “Genetically Modified (GM)”. This simple thing will make the reading easier for potential and future readers.

Line 32. Change the number “4” by letters “four”.

Line 34. When you write an acronym you must write in capitals the letters that you use to build it, for example, in the abstract you must write as follows: “Herbicide Resitant (HR)”. This simple thing will make the reading easier for potential and future readers.

Line 39. You have forgotten the point at the end of the last author of Glycine soja: “Zucc.”.

Line 41. You have to write “Glycine” in italics.

Line 75. You must write in brackets the meaning of each acronym in the text.

You must write explicitly the objectives of this manuscript.

Line 97. You have forgotten to write the point at the end of the Figure 1.

Line 115. You have forgotten to write the point at the end of the Figure 2.

Line 132. You have forgotten to write the point at the end of the Figure 3.

Line 158. You have forgotten to write the point at the end of the Figure 4.

Line 176. You have forgotten to write the point at the end of the Figure 5.

Line 199. You have forgotten to write the point at the end of the Figure 6.

Line 216. You have forgotten to write the point at the end of the Figure 7.

Line 233. You have forgotten to write the point at the end of the Figure 8.

In Materials and methods it would be very useful a distribution map of the wild soybean.

Otherwise, the authors adequately developed the Introduction, presenting the problems but you must write explicitly the objectives of this paper.

The methods are adequate.

The Discussion is well developed and the data presented are correctly compared with other papers.

The authors are to be congratulated for the results obtained in this article.

The English is good.

Reviewer 2 Report

In the present study, the fitness of F2 and F3 genotypes, derived from crosses of HR soybean line T14R1251-70 and two wild soybean accessions, LNTL and JLBC, were investigated under two soil conditions and with or without weed competition. In addition, the hard seeded ness of hybrid descendants was assessed by observing the seed coat structure and deter mining the emergence rate after burying in different soil depths for different lengths of time. The manuscript have several critical issues. The results section is not well presented and the key results are not interpreted and are general. In addition, the caption of the figures and the figure itself are vague and incomplete. Also, the results could not cover the aim of the work. Minor comments include:

-          In the caption of figure 2, parts A and B should be described.

-          Lines 123-125: this part needs to rewrite.

-          In the caption of figure 3, parts A and B should be described.

-          Line 139: replace “3” with “three”

-          In the caption of figure 5, parts A and B should be described.

-          In the caption of figure 6, parts A, B, C, and D should be described.

-          In figure 7, all elements such MW, PW, PF, etc. must be explained in caption. Please apply it in all figure captions.

-          Many parts of the discussion are not related to the results of the work.

Round 2

Reviewer 2 Report

The current version is improved.